# The Road to Practical Application of Cadmium Phytoremediation Using Rice

**DOI:** 10.3390/plants10091926

**Published:** 2021-09-15

**Authors:** Ryuichi Takahashi, Masashi Ito, Tomohiko Kawamoto

**Affiliations:** Akita Prefectural Agricultural Experiment Station, 34-1, Genpachizawa, Yuwa-Aikawa, Akita 010-1231, Japan; Itou-Masashi@pref.akita.lg.jp (M.I.); Kawamoto-Tomohiko@pref.akita.lg.jp (T.K.)

**Keywords:** cadmium, paddy field, phytoremediation, quantitative trait locus (QTL), rice, transporter

## Abstract

Cadmium (Cd) is a toxic heavy metal that causes severe health issues in humans. Cd accumulates in the human body when foods produced in Cd-contaminated fields are eaten. Therefore, soil remediation of contaminated fields is necessary to provide safe foods. Rice is one of the primary candidates for phytoremediation. There is a genotypic variation of Cd concentration in the shoots and grains of rice. Using the world rice core collection, ‘Jarjan’, ‘Anjana Dhan’, and ‘Cho-ko-koku’ were observed with a significantly higher level of Cd accumulation in the shoots and grains. Moreover, *OsHMA3*, a heavy metal transporter, was identified as a responsive gene of quantitative trait locus (QTL) for high Cd concentration in the shoots of these three varieties likewise. However, it is difficult to apply practical phytoremediation to these varieties because of their unfavorable agricultural traits, such as shatter and easily lodged. New rice varieties and lines were bred for Cd phytoremediation using *OsHMA3* as a DNA marker selection. All of them accumulated Cd in the shoots equal to or higher than ‘Cho-ko-koku’ with improved cultivation traits. Therefore, they can be used for practical Cd phytoremediation.

## 1. Introduction

Cadmium (Cd) is a toxic heavy metal that causes severe health issues in humans [1]. Cd may accumulate in the human body when it enters the food chain. The Codex Alimentarius Commission established a Cd limit in some agricultural products to provide safe food for human intake, e.g., 0.4 mg kg^−1^ for polished rice [2]. However, agricultural products harvested from Cd-contaminated fields can lead to Cd levels that exceed the Cd limit [3,4,5]. This high level of Cd can become a significant problem. Cd pollution of agricultural fields is mainly caused by the influx of Cd filled wastewater from mines and factories into agricultural fields. Cd pollution has an especially large impact on paddy fields. In addition, rice is the primary source of dietary Cd intake among Asians [3,4,5]. Therefore, soil remediation of contaminated fields, especially paddy fields, is necessary to provide safe foods for human health.

Phytoremediation is a soil remediation method that removes pollutants using plants. Phytoremediation is superior to other methods regarding its low cost and its ability to effectively purify large areas of field without any facilities. Generally, plants with large biomass and plants that accumulate high Cd levels in the aerial parts (shoots and grain) are required for effective phytoextraction. Cd hyperaccumulators, such as *Solanum nigrum*, *Pterocypsela laciniata*, *Sedum plumbizincicola*, are some of the candidates for soil remediation because of the high levels of Cd accumulation in the aerial parts [6,7,8]. However, most Cd hyperaccumulator have a small biomass which makes harvesting work difficult. Furthermore, a practical problem is the concern that they will spread as weed in the agricultural field after phytoremediation. On the other hand, some particular plant species, such as *Vigna unguiculata*, *Solanum melonaena*, *Momordica charantia*, *Nicotiana tabacum*, have also been reported for use with Cd phytoremediation [9]. In addition to these plants, rice (*Oryza sativa* L.) is also one of the primary candidates for soil remediation [10]. There is a genotypic variation of concentration of Cd ions in the shoots and rice grains, and some rice varieties showed a significantly high Cd accumulation in the shoots [11,12,13]. In addition, rice has large biomass. Furthermore, in the case of rice, there is a fully established cultivation and harvesting method, which means farmers can easily continue to work on soil remediation as they always have when cultivating and harvesting rice. Therefore, it is possible to perform phytoremediation by selecting high Cd-accumulating varieties to be cultivated without any special method or equipment.

In this review, we focus on Cd phytoremediation of paddy fields using rice. We discuss recent progress in breeding new rice varieties and future prospects for practical application.

## 2. Phytoremediation Using Local Rice Varieties

Among rice species, *Oryza sativa* subsp. *indica* and hybrid varieties of *O. sativa* subsp. *japonica* and *indica* (*japonica*-*indica*) exhibit a relatively high Cd accumulation in the shoots compared to the *O. sativa* subsp. *japonica* varieties [14]. Therefore, the candidate varieties for Cd phytoremediation were initially selected among *indica* or *japonica*-*indica* hybrid varieties. ‘Milyang 23’ is a *japonica*-*indica* variety that exhibits a relatively high Cd accumulation in the shoots [11]. When ‘Milyang 23’ was grown in Cd contaminated soil, it accumulated 10–15% of soil Cd in its shoots, and the decrease in Cd concentration in the soil after cultivation was the largest among the major crops, including the *japonica* rice variety ‘Nipponbare’ [15]. The Cd contents of soybean seeds cultivated in the same field after ‘Milyang 23’ was grown were less than those cultivated using untreated soil [16]. In a field experiment, the *indica* variety ‘Moretsu’ exhibited approximately 2.5-fold higher Cd accumulation than ‘Milyang 23’, and ‘IR-8’ (*indica* variety) also showed a high level of Cd accumulation in the shoots [17]. After performing phytoremediation using these two varieties for two years, the Cd concentration in the soil decreased by 18% compared with the soil before cultivation. Furthermore, the Cd contents in the grains of subsequently grown *japonica* rice were lower than those grown in a field without phytoremediation [17]. These results indicate that remediation using high Cd-accumulating rice is effective not only in the paddy field but also in the converted upland field.

The world rice core collection (WRC) covered the genetic diversity of 32,000 genotypes of cultivated rice [18]. Within WRC, ‘Jarjan’ ‘Anjana Dhan’, and ‘Cho-ko-koku’ accumulated a significantly high level of Cd in the shoots and grains [13]. Cd uptake by ‘Cho-ko-koku’ was higher than ‘IR-8’ even in the field, and the Cd concentration in the field soil and grains of subsequent cultivars decreased more than those in the control field after the cultivation of ‘Cho-ko-koku’ for 2 or 4 years [19,20]. Phytoremediation capacity of Cd removal from soil depends on Cd concentration in the aerial parts and biomass of the plants. ‘Jarjan’, ‘Anjana Dhan’, and ‘Cho-ko-koku’ have a large biomass, and these varieties were considered advantageous for phytoremediation. However, ‘Jarjan’, ‘Anjana Dhan’, and ‘Cho-ko-koku’ presented some difficulties in regard to practical phytoremediation because of their unfavorable agricultural traits, such as shatter and easily lodged. Therefore, it was necessary to breed new rice varieties, which accumulate high levels of Cd like ‘Jarjan’, ‘Anjana Dhan’, and ‘Cho-ko-koku’.

## 3. OsHMA3 Is an Important Metal Transporter for Cd Accumulation in the Shoots

Many studies have been done to identify and functionally analyze the genes related to Cd uptake and translocation in rice [21,22,23]. A primary quantitative trait locus (QTL) for increasing Cd concentration in the shoots and rice grains was located on chromosome 7 [24,25,26]. Backcrossed inbred lines (BILs) containing the QTL region (*qCdp7*) of ‘Jarjan’ allele reduced Cd concentration in the soil and subsequently cultivated rice shoots [27].

Heavy metal ATPase (HMA) is a metal transporter family, and some HMA in rice transports Cd [21,22,28]. OsHMA3 is localized to the vacuole membrane and functions as sequestration of Cd into vacuoles in the root cells [29,30]. One amino acid substitution causes a loss of the function of OsHMA3, and a disruption of the Cd sequestration into vacuoles leads to an increased Cd concentration in the cytoplasm (Figure 1). As a result, more Cd loads into the xylem and translocates from the roots to the shoots [31,32]. OsHMA3 from ‘Jarjan’, ‘Anjana Dhan’, and ‘Cho-ko-koku’ were observed with the amino acid substitution. Furthermore, the region detected *qCdp7* included *OsHMA3*, and *OsHMA3* was identified as a responsive gene of QTL for high Cd concentration in the shoots of ‘Jarjan’, ‘Anjana Dhan’, and ‘Cho-ko-koku’ [29,30,33]. Recently, it has been reported that various mutations of *OsHMA3* contribute to various levels of Cd accumulation in the shoots and rice grains [34,35,36]. These results suggest that OsHMA3 plays a vital role in Cd accumulation in the shoots and rice grains. In addition, the discovery of significant genes involved in Cd translocation from the roots to the shoots and Cd accumulation in the shoots enabled the efficient breeding of rice varieties for Cd phytoremediation.

## 4. Breeding New Rice Varieties for Phytoremediation

Rice line ‘MJ3’ and ‘MA22’ were obtained by gamma-ray mutation of ‘Jarjan’ and ‘Anjana Dhan’, respectively [37]. ‘MJ3’ and ‘MA22’ showed the same level of Cd extraction as ‘Jarjan’ and ‘Anjana Dhan’, respectively, with a non-shattering habit. However, the culm length of ‘MJ3’ was shorter than ‘Jarjan,’ and the lodging resistance of ‘MJ3’ was improved when compared with ‘Jarjan.’ On the other hand, ‘MA22’ showed almost the same culm length and was as easily lodged as ‘Anjana Dhan’.

‘TJTT8’ was developed from BILs derived from ‘Jarjan’ and ‘Tachisugata’ [38]. ‘Tachisugata’ is a *japonica*-*indica* hybrid variety used as a livestock feed. It shows a large biomass and lodging resistance because of its thick and rigid culms [39]. The grains of ‘TJTT8’ are dark brown color and can easily be distinguished from the grains of general *japonica* varieties like ‘MJ3’ (Table 1). ‘TJTT8’ was selected using the *qCdp7* allele and showed the same level of Cd extraction as ‘Jarjan’ at several locations of Cd-contaminated paddy fields in Japan [38]. However, the heading date and maturing date of ‘MJ3’, ‘MA22’, and ‘TJTT8’ were too late in northern parts of Japan. Late heading date and maturing date characteristics may lead to difficulty in the harvest work due to insufficient drying of the aerial parts and the need for obtaining seeds for subsequent planting. Therefore, other varieties suitable for the cultivation conditions in northern parts of Japan were required.

‘Akita 110’ is a rice line developed by the Akita Prefectural Agricultural Experiment Station located in the northern part of Japan. ‘Akita 110’ was selected from a cross between ‘Cho-ko-koku’ and ‘Akita 63’ [40]. ‘Akita 63’ showed lodging resistance and large biomass of the aerial parts [41]. In the process of breeding ‘Akita 110’, *OsHMA3* was used as a DNA marker to select plants that possessed this ‘Cho-ko-koku’ allele. As a result, the Cd extraction of ‘Akita 110’ was almost the same level as ‘Cho-ko-koku’. In one year of a large-scale field trial, soil Cd concentrations in plots remediated with ‘Akita 110’ were reduced by 15.5%, whereas remediation with ‘Cho-ko-koku’ reduced soil Cd levels by 10.1% [40]. However, the Cd extraction of ‘Akita 110’ was sometimes lower than that of ‘Cho-ko-koku’, depending on the field conditions. Then, a new rice line, ‘Akita 119’, was developed to improve stable Cd accumulation in the aerial parts. ‘Akita 119’ was obtained by a soft X-ray mutation of ‘Cho-ko-koku’ [42]. ‘Akita 119’ was also selected by the *OsHMA3* allele of ‘Cho-ko-koku’. The culm length of ‘Akita 119’ was around 30 cm shorter than ‘Cho-ko-koku’, and the lodging resistance of ‘Akita 119’ was improved compared to ‘Cho-ko-koku’. ‘Akita 119’ had many panicles, and its biomass was as great as that of ‘Cho-ko-koku’, even though the culm was short. The grains of ‘Akita 119’ are slender with a light brown color (Table 2). The grains can also be distinguished from the grains of the general *japonica* varieties and do not mix in the distribution process if separated by a sieve.

**Table 1 plants-10-01926-t001:** Characterization of ‘MJ3’, ‘TJTT8’, ‘Akita 110’, and ‘Akita 119’. “2*Tachisugata” means crossing twice with ‘Tachisugata.’

	MJ3	TJTT8	Akita 110	Akita 119
Crossing combination (Mother/Father)	Gamma-ray mutation of ‘Jarjan’	Tachisugata/Jarjan //2*Tachisugata	Cho-ko-koku/Akita 63	Soft X-ray mutation of ‘Cho-ko-koku’
Lodging resistance	Weak to medium	Very strong	Medium	Weak to medium
Shattering resistance	Strong	Strong	Strong	Strong
Distinguish points with general *japonica* varieties	Long culm length Grain shape Grain color	Long culm length Grain shape Grain color	Grain shape Grain color	Many panicle number Grain shape Grain color
Reference	[37]	[38]	[40]	[42]

When ‘Akita 119’ was cultivated in a Cd-contaminated paddy field in Akita Prefecture, the Cd extraction of ‘Akita 119’ was almost the same level as ‘Cho-ko-koku’ in all experiment fields from 2011 to 2016, whereas ‘Akita 110’ was lower than ‘Cho-ko-koku’ in some fields (Table 2). We also cultivated ‘MJ3’, ‘MA22’, and ‘TJTT8’ in the same Cd-contaminated paddy field in Akita Prefecture from 2014 to 2016 (Table 2). ‘MA22’ showed the highest Cd extraction among trial plants in 2014 but lodged more severely than ‘Cho-ko-koku.’ On the other hand, Cd extraction of ‘MJ3’ and ‘TJTT8’ was about 1.5 times higher than that of ‘Cho-ko-koku’ on average, although some cultivating problems of these lines, such as late heading and maturing date, were revealed in northern parts of Japan.

## 5. Another Approach to Breeding New Rice Varieties

Morphological improvements are another approach to breeding new rice varieties for phytoremediation. They promise a further increase of biomass in the aerial parts by crossing varieties with large biomass, such as rice used for livestock feed. On the other hand, expanding the rhizosphere might also be an effective strategy. It is considered that deeper rooting and maximal root length enhance nutrition and water uptake from soil [43,44]. Similarly, root modification may enhance Cd absorption of phytoremediation rice.

In general, as the dry weight in the shoots increases, Cd concentration in the shoots decreases. Therefore, it is necessary to improve Cd translocation efficiency from roots to shoots to accumulate a higher Cd level in the aerial parts. OsNRAMP5 is a manganese, iron, and Cd transporter and is recognized as a major route of Cd influx into root cells [45,46]. Mutant rice with defective OsNRAMP5 showed little absorption of Cd [47]. On the other hand, rice, whose gene expression of *OsNRAMP5* was suppressed by RNAi, accumulated higher amounts of Cd in their shoots than the wild type, whereas the total Cd content (roots plus shoots) was reduced [45]. This is considered to be because the expression of genes involved in metal transport was enhanced. This indicates that combining some genes related to Cd absorption and translocation is effective in enhancing Cd accumulation in the aerial parts.

A major QTL and responsive gene for increasing Cd concentration in the shoots and the grains of rice was identified as *OsHMA3* within chromosome 7 [24,25]. In addition, other QTLs for Cd accumulation in the aerial parts and Cd translocation from roots to shoots have been detected in other studies [48,49,50] (Table 3). Thus, the mechanisms of Cd accumulation may be different among each high Cd-accumulating variety. Using these varieties in combination is effective for breeding new rice varieties for Cd phytoremediation (Figure 2).

## 6. Another Approach for Efficient Cd Phytoremediation

Cd absorption from soil to rice roots is maximized under drained and oxidative soil conditions [51]. This is because in such situations, Cd in the soil exists in a chemical form that is easily absorbed by the rice. Therefore, making more Cd in the soil into an available form affects the efficiency of phytoremediation. Some microorganisms, including bacteria and arbuscular mycorrhizal fungi, can convert unavailable Cd in the soil, facilitating its bioavailability for plants. In addition, they are well known to be able to increase the tolerance of plants to Cd stress. Numerous microbes have been reported to enhance the efficiency of Cd phytoremediation by some plant species [52,53,54]. In many cases, microorganisms interact with a limited host plant, so it is necessary to isolate microbes that operate under rice cultivation conditions.

## 7. Future Prospects and Conclusions

In addition to *OsHMA3* and *OsNRAMP5*, many genes related to Cd uptake and translocation are well known in rice. For example, *OsIRT1*, *OsIRT2*, and *OsNRAMP1* are involved in Cd uptake and Cd translocation within plants [55,56,57], and *OsHMA2* plays a role in Cd translocation from roots to shoots and/or Cd distribution in the nodes [58,59,60]. Furthermore, *OsZIP1* functions as a metal exporter under Cd excess condition to prevent Cd stress [61]. Therefore, transgenic rice development using these gene is one of the most efficient methods for breeding a new rice variety for Cd phytoremediation. OsMTP1, which is localized to the vacuole membrane, functions as sequestration of Cd into vacuoles, and overexpressing *OsMTP1* enhanced Cd accumulation in transgenic tobacco [62,63]. Higher accumulation in the shoots was also observed in the *OsNRAMP5*-RNAi rice of ‘Anjana Dhan’, even in a Cd-contaminated field [64]. However, transgenic rice has not been practicalized in any country, even in the inedible varieties. If genetically modified crops are commercialized, it is expected that the development of new rice varieties for Cd phytoremediation will also progress dramatically.

Although a limit on Cd content in crops has been established, it is not clear how much the Cd concentration in the soil should be reduced by phytoremediation for the safe production of subsequent crops. Clarification of the change of soil Cd concentrations before and after phytoremediation is complex because it strongly depends on weather and soil conditions. Furthermore, Cd-contaminated fields, which can be used to perform field trials for phytoremediation, are limited. However, more field trial results are necessary to establish a guideline for the end of phytoremediation.

Phytoremediation is an effective method to restore Cd-contaminated soil to an agricultural field. For further amplification of Cd phytoremediation, it is necessary to establish an efficient method that also reduces the burden on farmers. It must include field work and pecuniary, because farmers usually have no income during the phytoremediation process. Therefore, it is necessary that researchers and government work together toward the practical application of Cd phytoremediation.

## Figures and Tables

**Figure 1 plants-10-01926-f001:**
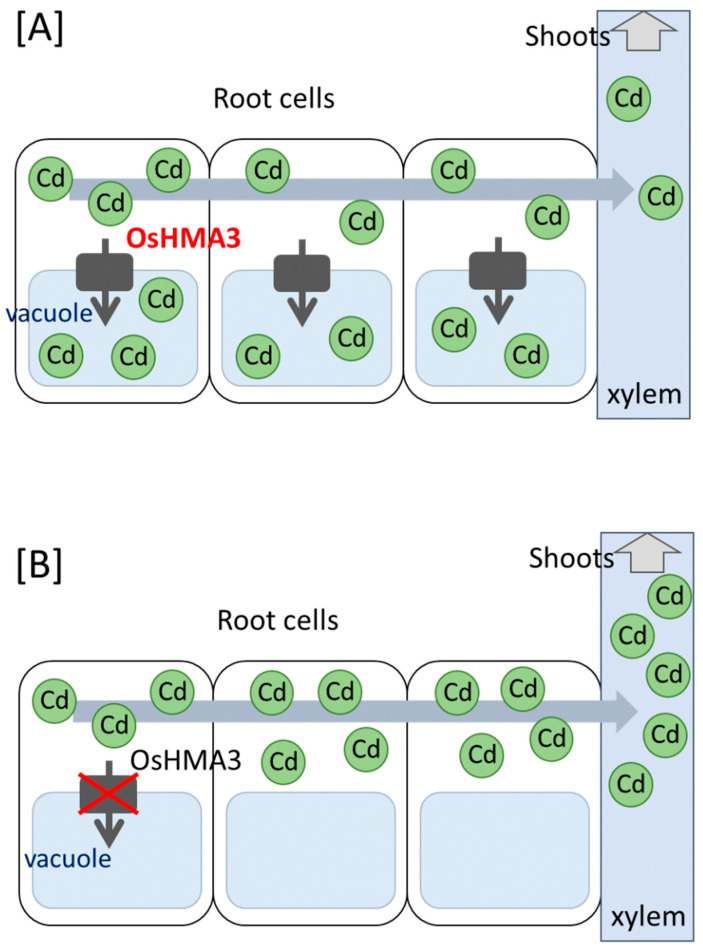
Proposed model for the role of OsHMA3 in Cd translocation from the roots to the shoots. The Cd is sequestered into the vacuole via OsHMA3 in normal varieties (**A**), whereas not in high Cd-accumulating varieties (**B**). The remaining Cd in the cytoplasm is loaded into the xylem.

**Figure 2 plants-10-01926-f002:**
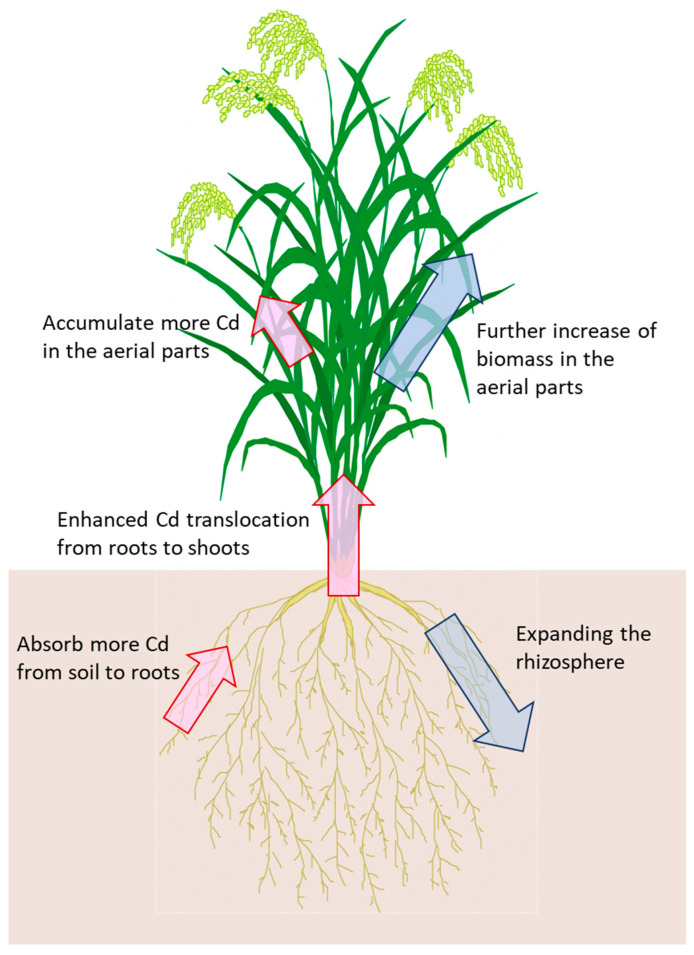
The breeding goals for new rice varieties of Cd phytoremediation. Blue arrows and red arrows indicate morphological and physiological improvements, respectively.

**Table 2 plants-10-01926-t002:** Cd extraction ratio. Each variety or line was cultivated in Cd contaminated field in Akita Prefecture. The ratio is calculated by the Cd extraction of ‘Cho-ko-koku’ cultivated in the same field as 1.00. “-” indicates no data.

	Cd Extraction Ratio (Ratio with ‘Cho-ko-koku’ as 1.00)										Reference
Year	2011	2012	2013	2014		2015		2016		Average	
Soil Cd Conc. (mg/kg)	0.55	1.47	0.73	0.96	0.96	0.62	0.75	0.51	0.55		
Akita 119	1.10	1.02	1.23	1.06	0.92	1.09	1.16	0.89	1.00	1.05	[42]
Akita 110	1.18	0.71	0.62	1.34	1.04	0.67	0.65	-	0.99	0.90	[40]
MJ3	-	-	-	1.38	-	1.58	-	-	-	1.48	[37]
MA22	-	-	-	1.64	-	-	-	-	-	1.64	[37]
TJTT8	-	-	-	1.35	-	1.64	-	1.43	-	1.47	[38]
Cho-ko-koku	1.00	1.00	1.00	1.00	1.00	1.00	1.00	1.00	1.00	1.00	[13,26]

**Table 3 plants-10-01926-t003:** QTL region and the function reported previously.

Function	Crossing Parents		ChromosomalRegion of QTL	Reference
	High-Cd	Low-Cd		
Cd translocation from root to shoot	Anjana Dhan	Nipponbare	Chr. 7	[24]
High Cd concentration in grain	Habataki	Sasanishiki	Chr. 7	[25]
High Cd concentration in grain	Kasalath	Koshihikari	Chr. 6	[48]
Low Cd concentration in grain	Kasalath	Koshihikari	Chr. 3, 8	[48]
Cd translocation from root to aerial part	Kasalath	Nipponbare	Chr. 4, 11	[49]
Cd translocation from root to shoot	Badari Dhan	Shwe War	Chr. 11	[50]

## Data Availability

No supporting data in this study.

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
