# Peer review of "The Road to Practical Application of Cadmium Phytoremediation Using Rice"

_plants, 2021, doi:10.3390/plants10091926_

Round 1

Reviewer 1 Report

The researchers review the application of rice as a bioremediation in cadmium fields contamination, as well as the genetic improvements in rice to create new varieties.

Even when the subject is not new, cause the first studies date from the end of the sixties, the studies of the genes are most recent, so this reviews focus on literature from the eighties and so on.

This article contributes to the model of the movement of cadmium from the roots to the shoots and rice grains.

All researchers have experience in this topic, so it contributes to science and the journal.

The paper is well written; however, some wordy sentences can be change

The text is comprehensible, but sometimes it is not necessary to repeat the names of the varieties in the same sentence.

It will be nice if, in the beginning, you write the Genus and species. Oryza sativa subsp. Japonica; O. sativa subsp. Indica

They conclude well based on the review done to the subject

Reviewer 2 Report

The review is well constructed and discusses several aspects involved in cadmium accumulation in rice plants. I found the exploration of metal transporters role in Cd uptake and translocation very interesting. In addition, cited reference are of interest. However, I wonder why use edible species for phytoremediation is of concern? One of the major issue is the control of cadmium accumulation in rice grains to reduce the potential health risk for the populations with a rice-based diet. Why is there a growing scientific literature exploring the potential of rice plants in phytoremediation? In my opinion, the authors should argue about the reasons why phytoremediation based on rice plants rather than others would be suitable or preferable. For example, the authors indicate in the review that after performing phytoremediation using these indica and japonica-indica for two years, the Cd concentration in the soil decreased by 18%. Is this choice sustainable and effective? It would mean (i) taking land away from cultivation for human consumption for long periods with a marginal reduction in cadmium levels in soils, (ii) and producing contaminated food (is it then discarded)? Could other cadmium hyperaccumulating plant species be used to solve the problem more efficiently? I am thinking, for example, of non-edible species capable of colonizing soils with high cadmium content. I also suggest minor review to be addressed as follows:

  • As concerns the sentence “Cd accumulates in the human body when foods produced in Cd-contaminated fields are eaten”, it would be more correct to write that Cd may accumulate in the human body when it enter in the food chain. It is not always the case that if an agricultural product is grown on contaminated soil, it is also contaminated. Plants can implement exclusion strategies to prevent the potentially toxic element accumulation within them.
  • Regarding “Rice is one of the primary 10 candidates for phytoremediation.” Is it really? I suggest the authors the manuscript Raza et al., 2020 (doi: 10.3390/biology9070177) and reference therein.
  • I invite the authors to review the abstract since there are identical phrases to those used in the body of the manuscript (e.g. lines 25-26).
  • Lines 25-26: Please, refer to the first comment.
  • Lines 27- 29: Please, specify which is the concentration limit for Cd in agricultural products you are referring to.
  • Lines 29-31: Please add references which provides evidences for high Cd levels in these products.
  • The authors should specify the aims of the review in the introduction section.
  • Line 80: Reference Ishikawa et al. should be adjusted.
  • Lines 120-124: Please insert the reference.
  • As concerns the role of metal transporter, is the role of Zn transporter proteins (ZRT) recognized for rice?
  • Lines 185-187: I find this part quite out of context.
  • In my opinion, the authors should explore (whether possible) the role of plant-microbes interaction in Cd phytoremediation as future perspective. Microorganisms (both bacteria and arbuscular mycorrhizal fungi) may be able to mobilize unavailable Cd in soil, facilitating its bioavailability for plants. In addition, they are well known to be able to increase the tolerance of plants to Cd stress.

Reviewer 3 Report

The manuscript entitled "The road to the practical application of cadmium phytoremediation using rice" is a concise review article that corresponds well with the topic outlined in its title. I am only unsatisfied with the last (missing) paragraph of the introduction, which should contain information about the issues that the authors will then discuss in the manuscript and on what basis they chose the publications cited in the text. This will make it easier for less experienced readers to trace the text. Also, only in the introduction, I proposed alternative terms to avoid the term "phytormediation" multiplied in the following lines (see corrections in the manuscript).

The only concern is whether the biomass from the varieties / lines used in the cleaning of cadmium-contaminated fields will not be used to feed animals or used (grain) as food for poor people. But this is a separate issue, not affecting the high value of the manuscript under review.

Author Response

请参阅附件。
